# Uncalibrated Models Can Improve Human-AI Collaboration

**Kailas Vodrahalli**
Stanford University
kailasv@stanford.edu

**Tobias Gerstenberg**
Stanford University
gerstenberg@stanford.edu

**James Zou**
Stanford University
jamesz@stanford.edu

## Abstract

In many practical applications of AI, an AI model is used as a decision aid for human users. The AI provides advice that a human (sometimes) incorporates into their decision-making process. The AI advice is often presented with some measure of "confidence" that the human can use to calibrate how much they depend on or trust the advice. In this paper, we present an initial exploration that suggests showing AI models as more confident than they actually are, even when the original AI is well-calibrated, can improve human-AI performance (measured as the accuracy and confidence of the human's final prediction after seeing the AI advice). We first train a model to predict human incorporation of AI advice using data from thousands of human-AI interactions. This enables us to explicitly estimate how to transform the AI's prediction confidence, making the AI *un*calibrated, in order to improve the final human prediction. We empirically validate our results across four different tasks—dealing with images, text and tabular data—involving hundreds of human participants. We further support our findings with simulation analysis. Our findings suggest the importance of jointly optimizing the human-AI system as opposed to the standard paradigm of optimizing the AI model alone.

## 1 Introduction

In safety-critical settings like medicine, AI is often integrated as interactive feedback, allowing a human to decide when and to what extent the AI's "advice" is utilized [29]. For example, in medical diagnosis tasks, this places the AI in a similar category as lab tests or other exams a doctor may order to aid in diagnosis. This mitigates the often black box nature of AI algorithms that limit a user's trust and usage of AI [6, 24, 34, 18].

Typically, these algorithms are designed and optimized to minimize loss using standard training objectives independent of the human end user. In this paper, we question this premise and ask whether a joint optimization of the entire human-AI system is possible. In particular, we revisit the conventional wisdom that models with calibrated confidence enable more accurate transfer of prediction uncertainty between models and/or humans [8]. We propose calibrating a model for human use, with the intuition that humans are often unable to accurately integrate calibration information correctly and so may benefit from uncalibrated information. This approach is complementary to existing ideas on how to help humans utilize AI advice (e.g., through some form of explanation accompanying the advice or through feedback on model/human performance) [36, 14].

We draw on prior work modeling human predictions [30] and assume black box access to an AI algorithm which provides advice. From data collected on thousands of human interactions, we fit a model for how humans incorporate (or ignore) AI advice. Using this learned human behavior model, we optimize a monotonic transformation that modifies the AI's reported confidence to maximize human-AI system performance (measured as the human's final accuracy). We demonstrate first in simulations and subsequently with real-world data using crowd-sourced human study participants

36th Conference on Neural Information Processing Systems (NeurIPS 2022).

that our modified AI advice results in higher overall performance. These results suggest there is merit to jointly optimizing human-AI systems and that an optimal AI algorithm may not be optimal for human use.

**Our contributions** We propose and empirically validate a framework for optimizing an AI algorithm with respect to the human end user, resulting in *human-calibrated AI* that calibrates AI models for human use. We only require black box access to the AI. Our framework relies on a model for human behavior, which we create using a large collection of empirical data on human interaction with AI advice. To the best of our knowledge, these are some of the first results using an empirically validated model for human behavior to optimize AI performance in a collaborative human-AI system. We would like to emphasize, however, that this these methods are not currently suitable for practical use.

**Ethical considerations** One of the main conclusions of our study is that human users do not consider confidence in the "optimal" way (i.e., how we model confidence in statistics). In an attempt to correct for this, we present both empirical and simulation-based evidence that modifying the presented AI advice improves overall performance. One potential question about our approach is: does modifying the AI's confidence constitute misleading the user? We agree – in our experiments, the AI can mislead the user. However, our goal here is not to propose a method that should be used in practical applications, and rather to highlight the importance of modeling human interaction when designing AI for human use.

**Related works** The experimental setup we use comes from the advice utilization community. Here, studies often use the JAS framework for measuring and comparing the effect of different forms of advice [28]. In many of these studies, algorithmic advice is manipulated to elicit various responses from humans. We take this a step further in our paper by manipulating the advice to improve model calibration with respect to the user and improve performance. Previous work has studied the effect of peer vs. expert advice [15], automated vs. human advice [22, 5, 15, 21], task difficulty [7], and advice confidence [25, 26]. Some work has also sought to characterize the perceptions people have of an AI when making decisions with AI advice and how these perceptions affect trust and utilization of the AI [13]. Of particular relevance to our work is [25], which provides evidence that uncalibrated, confident advice is more utilized by humans in certain settings. We also leverage a proposed model for human behavior from [30].

The conventional approach to human-AI collaboration is to ensure model and human calibration [8, 3] and to make the AI's predictions explainable [31, 27]. These explanations can affect human understanding of AI confidence [12, 35, 36, 14]. It is also standard practice to optimize the AI in isolation (i.e., maximize the AI performance). However, recent work suggests that the objectives used to optimize the AI do not adequately consider the human-AI team. Alternative approaches include learning to defer, where a classifier determines when expert input is required [16, 19]; learning to complement, where the AI is optimized to perform well on only the tasks that humans struggle with [33]; and methods that optimize objectives with costs assigned to requesting AI advice and/or human expert input utilization [1]. Our work is most related to [1], which also emphasizes the importance of confident AI predictions for human decisions. Our approach is complementary to these prior works and contains several novel aspects. We model human behavior using empirical data, demonstrate that this human model allows us to optimize the AI advice, and demonstrate both in simulation and empirically that our modified advice improves overall system performance. Furthermore, we only require black-box access to the AI advice. In relation to [1], we further differentiate our work as the model for human interaction is different – we give advice after the user makes an initial assessment.

## 2  Experimental setup

Our goal is to augment the performance of a human-AI system by optimizing the AI for use by a human. To make this goal tractable, we assume (1) that we have a model for human interaction with AI advice, and (2) that we have black box access to the AI algorithm producing the advice.

**Human-AI interaction**

The human-AI interaction involves two stages. First, a human is shown a task and completes it. AI advice is then presented, and the human is allowed to modify their response. As we measure performance before and after receiving advice, this model is conducive to studying the effect of advice and so sees common usage in the literature [22, 30, 17].

**Data collection and tasks**

We collected data from crowd-sourced participants on a diverse set of tasks that all have the same basic design. We focused on binary prediction tasks where humans respond on a continuous scale. The use of the continuous scale allows for a more rich analysis of the user's prediction than just the simple binary prediction label would permit.

Participants were shown a sliding scale with each end indicating a different binary response. They were instructed to move a pointer on the scale to the location that indicated their confidence and their response, with the midpoint on the scale indicating an uncertain response. Participants submitted an initial response; subsequently, the advice was indicated with a visual marker on the scale and participants were allowed to adjust their initial response. Participants were incentivized so that higher confidence in correct and incorrect answers increased and decreased their monetary reward respectively.

We collected data on a set of four tasks from diverse data modalities. These tasks were originally proposed in prior work [30]. The tasks were designed to be accessible to the general public to allow use of crowd-sourced study participants for obtaining empirical data. See below for a brief description of the tasks we used; examples of task instances from each of the four tasks are shown in Appendix B.2. The data collected is summarized in Table 1. Each task consists of 32 binary questions. For each task, the 32 questions were carefully selected to have a range of difficulty. More details on the distribution of task difficulty is presented in Appendix B.3.

All participants received the same set of questions, though the order was randomized for each participant. All participants also received the same advice, although a small amount of noise was randomly added for each participant. The advice had an accuracy of $\sim 80\%$ for each task; participants were informed of this accuracy. However, no feedback on AI or human accuracy on individual questions was provided during the experiment.

Since the datasets we are using are relatively small and curated primarily for human use rather than AI training, we opted to use aggregated human responses as a proxy for an AI model. In particular, the advice was constructed by averaging the responses of a prior group of human participants who responded to the questions without receiving any advice. This allowed us to ensure the advice given was "reasonable" in the sense that the advice is generally less confident on difficult tasks and more confident on easy tasks. We checked the calibration of this advice by computing the expected calibration error (ECE) [20]. The ECE for AI advice is $0.074$, indicating the advice is roughly calibrated (0 indicates perfect calibration).

We recruited 49-79 participants for each task, resulting in several thousand datapoints (each question is one datapoint). Participants are US residents, roughly 50% are female, and the average age is approximately 30 years old. Despite limitations in the age distribution and geographic location of participants, a survey question we asked indicated a varied range of AI familiarity in our participants.

We followed standard practice in collecting data. Our study protocol was approved by our institution's IRB board. The data collection process was low risk, and we ensured samples contained diverse participants across age, sex, and ethnicity. We compensated participants at $\geq$ \$10.00 per hour (depending on bonus pay). The total cost was around \$600 (Section 2) + \$750 (Section 5.2).

We have four types of data with corresponding user tasks. **Art dataset (Image data)** Contains images of paintings from 4 art periods: Renaissance, Baroque, Romanticism, and Modern Art. Participants were asked to determine the art period a painting is from given a binary choice. **Cities dataset (Image data)** This dataset contains images from 4 major US cities: San Francisco, Los Angeles, Chicago, and New York City. The task is to identify which city an image is from given a binary choice. **Sarcasm dataset (Text data)** This dataset is a subset of the Reddit sarcasm dataset [10], which includes text snippets from the discussion forum website, Reddit. Participants were asked to detect whether sarcasm was present in a given text snippet. **Census dataset (Tabular data)** This dataset comes from US census data [32]. The task is to identify an individual's income level given some of their demographic information—e.g. state of residence, age, education level.

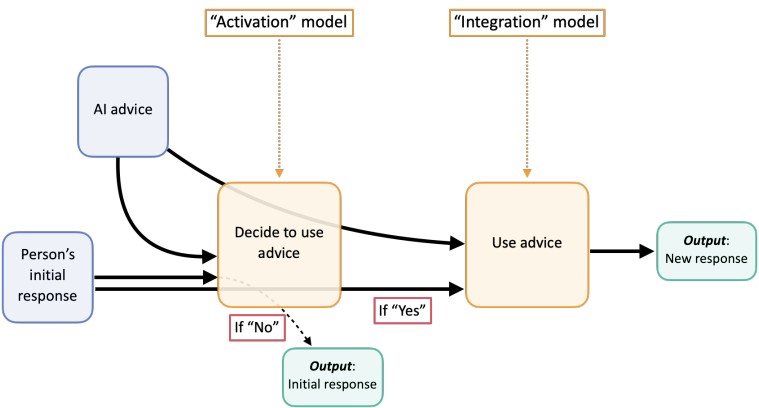

Figure 1: Activation-integration model for human behavior.

Table 1: Summary of data used for fitting our human behavior model. This data is publicly available at https://github.com/kailas-v/human-ai-interactions.

| Task | # Participants | # Observations | Activation Rate | Accuracy (before advice) | Accuracy (after advice) |
|---|---|---|---|---|---|
| Art | 68 | 2176 | 0.523 | 63.7% | 75.5% |
| Cities | 79 | 2528 | 0.557 | 70.7% | 77.1% |
| Sarcasm | 49 | 1567 | 0.347 | 72.7% | 76.1% |
| Census | 50 | 1600 | 0.475 | 70.1% | 73.6% |

## 3 Human behavior model

Table 1 provides a summary of the data collected for developing our human behavior model. We find that humans actually changed their response after receiving advice roughly $50\%$ of the time, with the exact number shown in the "Activation Rate" column of Table 1. When a person does change their response, we call this person "activated." Specifically, a person is activated if their final response changes by at least a small amount.

This observation motivates a simple, two-stage model for human behavior: a human first decides whether to modify their response ("activation stage"), and subsequently, if they were activated, they modify their response ("integration stage"). A diagram of this model is shown in Figure 1. This two-stage model is consistent with previously proposed models from the psychology and HCI literature [9, 30].

We fit functions for each of the two stages: an activation model that predicts how likely it is that a person is activated, and an integration model that predicts how the person modifies their advice, assuming they are activated. Taken together, we can model human behavior in the two-stage human-AI system we use to run our experiments. We define this model's behavior more rigorously in Section 4. Our fitted models will only be used to optimize our advice-modifying function and will not be used a test-time. As such, we can utilize demographic information about the participants to fit the model, though such information may not be available during a real-world deployment. We opt to fit a small neural network for each of these stages, as we require a differentiable model (for easy optimization) with moderately more complexity than a linear model.

### 3.1 Activation model

Our activation model takes 12 input features extracted from a single person-task instance interaction. It outputs a probability for whether the person will be activated on the given task instance. The input features are based on the person's initial response, the AI advice, and several demographic features including age and sex. We do not encode the initial response or AI advice directly as the label (the

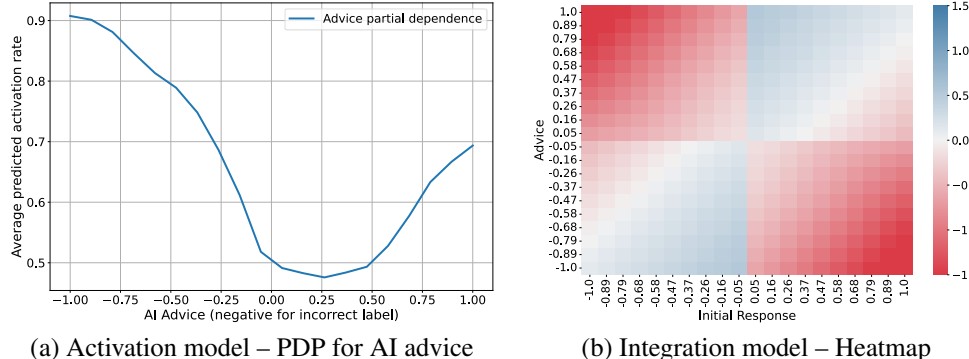

(a) Activation model – PDP for AI advice       (b) Integration model – Heatmap

Figure 2: Our trained model for human behavior: (a) Partial dependence plot measuring the effect of the advice on activation. (b) Heatmap of the average behaviour of our integration model for human behavior when we vary unmodified AI advice and the human's initial response.

left or right slider location of the correct answer) is randomized for each participant. Instead, we use the response and advice confidence, as well as a binary feature that encodes whether the response and advice agreed on the label. This implies a symmetry in the model's predictions. Please see Appendix D.1 for a description of all the features used.

We use a 3 layer neural network with ReLU non-linearities for the activation model. We also attempted to use other simple algorithms for modeling the activation behavior of humans, like linear models, but found the neural networks to perform better. We trained the network using binary cross-entropy loss with an Adam optimizer [11] and stopped training when validation loss increased, as is standard. The activation model achieves a ROC-AUC $> 0.81$ on held-out test participants, suggesting reasonable performance despite the high variability in our dataset.

In Figure 2a, we show a partial dependence plot for the learned activation model. This plot shows the average behavior (across our entire test dataset) of the activation model when we fix the AI advice to be a certain value (the x-value) for all data points. The AI advice can be positive (correct advice) or negative (incorrect advice), while the magnitude corresponds to the advice confidence. We note that (1) there is a base level of activation around 50% and (2) our activation model predicts people will be most activated when the advice is confidently incorrect. This occurs because (a) most people in our dataset get answers correct, on average, and seeing confident advice of the opposite label tends to make people change their initial response (e.g. by decreasing confidence even if the predicted label does not change). And because (b) confident advice that agrees with a person's initial response (whether correct or incorrect) tends to increase their confidence. We also observe that (3) the activation rate increase when advice is confident and correct for the same reason previously noted in (2b). The activation rate is lower for correct advice than for incorrect advice as most people in our dataset are, on average correct, which tends to invoke a smaller response.

## 3.2 Integration model

Our integration model is similar to our activation model, but optimized for a different function. In particular, we use the same input features, model architecture, and training procedure. The integration model outputs a prediction for $sign(r_1)(r_2 - r_1)$, where $r_1, r_2 \in [-1, 1]$ are the initial and final responses respectively. Note that $r_1, r_2$ are probabilities normalized to the $[-1, 1]$ scale. The training minimizes the RMSE between the predicted change and the actual change. The optimization is performed on only the subset of the dataset that was actually activated (roughly $50\%$ of the data). Our integration model has a reasonable performance, with an RMSE of $0.25$ (the maximum absolute error possible is 2) and $R^2$ of $0.73$ on test data.

In Figure 2b, we plot a heatmap of the fitted integration model's behavior. We plot the average output of our integration model across our test data split when the person's initial response (x-axis) and AI advice (y-axis) are fixed to a certain value for all datapoints. The sign and magnitude of the initial response and AI advice correspond to correctness (negative is incorrect) and confidence. The

integration model prediction can be positive (increasing confidence in the initial predicted label) or negative (decreasing confidence in the initial predicted label / change in the predicted label).

We observe three features in the heatmap: (1) The heat map is symmetric across quadrants 1, 3 and 2, 4. This is by design. As the model does not use the initial response and AI advice values directly – it uses their magnitudes (confidence) as well as a third, binary term indicating whether they agree on label – the integration model can only predict a delta change relative to the sign of the initial response. (2) The output has largest magnitude when the advice is confident and opposite the person's initial response. (3) If the advice and the person's initial response share the same label, the integration model output is largest when the person's initial response has low confidence and the model has high confidence. These observations confirm our integration model has reasonable operation behavior.

## 4  Modifying the AI advice

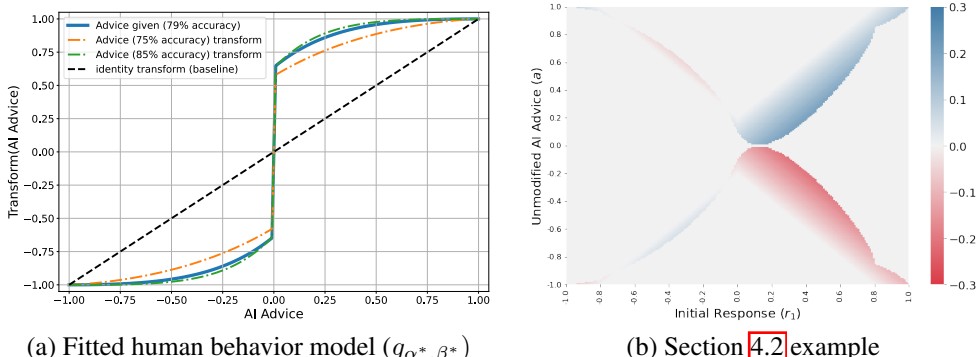

(a) Fitted human behavior model ($g_{\alpha^*, \beta^*}$)  (b) Section 4.2 example

Figure 3: Simulation results. (a) Fitted modified advice vs. original unmodified advice. We show results across synthetically varied average advice accuracy in the training data. The curve labeled "Advice given" is what we actually show people in our experiments. (b) Optimal human-calibrated AI output for the setting described in Section 4.2. Heatmap shows delta change in advice compared to calibrated advice: red and blue values indicate where the AI should be under- and overconfident respectively in order to achieve the best human-AI system output.

Here we discuss how we modify the raw AI advice output. First let's establish some notation. The notation described below is in reference to a single datapoint; later, we will add the subscript $i$ to index across datapoints.

Let $A \in \mathbb{R}$ denote the inverse sigmoid of the advice presented to participants (i.e., we present the AI advice as $\sigma(A)$). We will work with $A$ rather than $\sigma(A)$ here, as we will aim to replace the $\sigma$ transform with an optimized function that improves human-AI performance.

Let $g : \mathbb{R} \rightarrow [0, 1]$ be a function (which we will optimize) that maps the AI advice to a probability that the user is shown. Note that for the baseline, "unmodified advice," we set $g$ to be the sigmoid function, $\sigma(x) = \frac{1}{1+e^{-x}}$.

Let $r_1, r_2$ be the initial and final responses from a participant ($r_2$ is a function of $r_1$, $A$, and other features), and let $y \in \{0, 1\}$ denote the true label. Let $\mathbf{u}$ represent demographic features of a person, and let $\mathbf{x} = (r_1, A, \mathbf{u})$ be the input feature vector to $f_{\text{activation}}$ and $f_{\text{integration}}$, our activation and integration models for human behavior models described in Section 3. To simplify notation, we let $f_{\text{integration}}$ denote the function that outputs the predicted $r_2$ for an activated person (rather than the delta from $r_1$ that we optimize for). Finally, let $f_{\text{HB}}$ represent the entire model for human behavior. We assume all feature preprocessing is hidden inside $f_{\text{activation}}$ and $f_{\text{integration}}$ for convenience of notation.

### 4.1  Optimizing for human-AI performance

We consider $g$ of the form

$$g(A) = g_{\alpha, \beta}(A) = \frac{1}{1 + e^{-\text{sign}(A)(\alpha|A|+\beta)}}, \tag{1}$$

where $\alpha, \beta \in \mathbb{R}_{\geq 0}$. Note that setting $(\alpha, \beta) = (1, 0)$ results in $g_{1,0} = \sigma$, the sigmoid function used to produce the unmodified advice. Note that we do not exactly perform a linear transformation on $A$: the $|A|$ and $\text{sign}(A)$ terms are included to ensure that $g$ does not change the label recommended as it makes the function symmetric around $A = 0$. $\alpha$ modulates the rate at which the presented advice increases with AI confidence, while $\beta$ adjusts the minimum confidence level of the presented advice. For example, if $\beta = 1$, the presented confidence is $> \frac{e}{1+e} > 0.73$ (relative to the chosen label). In general, we can use other differentiable functions for $g$; we chose the form in Equation 1 because it is easy to optimize, simple to understand, and is still flexible enough to fit our data.

As a shorthand for when we modify the features input to $f_{\text{activate}}$ and $f_{\text{integrate}}$, we denote $g(\mathbf{x}) = (g(A), r_1, \mathbf{u})$. We will optimize for $\alpha, \beta$ to maximize the expected final accuracy using our human behavior model, $f_{HB}$. As discussed in Section 3, $f_{HB}$ has the following representation:

$$f_{HB}(g(\mathbf{x})) = f_{HB}(g_{\alpha,\beta}(\mathbf{x})) = \begin{cases} r_1 & \text{w.p. } 1 - f_{\text{activate}}(g_{\alpha,\beta}(\mathbf{x})) \\ f_{\text{integrate}}(g_{\alpha,\beta}(\mathbf{x})) & \text{w.p. } f_{\text{activate}}(g_{\alpha,\beta}(\mathbf{x})) \end{cases}$$

We then solve the following optimization problem to obtain $\alpha, \beta$:

$$(\alpha^*, \beta^*) = \arg\min_{\alpha,\beta} \mathbb{E}\left[L(f_{HB}(g(\mathbf{x})), y)\right], \tag{2}$$

where $L$ is the binary cross entropy loss function. The expectation is taken with respect to the randomness in $f_{HB}$ and the data distribution. Given training dataset $D$, we optimize the following expression:

$$\arg\min_{\alpha,\beta} \sum_{(\mathbf{x}_i, y_i) \in D} \mathbb{E}[L(f_{HB}(g(\mathbf{x}_i)), y_i)] =$$

$$\arg\min_{\alpha,\beta} \sum_{(\mathbf{x}_i, y_i) \in D} L(r_{1_i}, y_i) \cdot (1 - f_{\text{activate}}(g(\mathbf{x}_i))) + L(f_{\text{integrate}}(g(\mathbf{x}_i)), y_i) \cdot f_{\text{activate}}(g(\mathbf{x}_i))$$

In practice, we carry out this optimization using stochastic gradient descent, which is possible as we choose $f_{\text{activate}}$ and $f_{\text{integrate}}$ to be differentiable functions (see Appendix D.2). For data, we used the empirical data described in Section 2. The full training procedure is visualized in Figure 4.

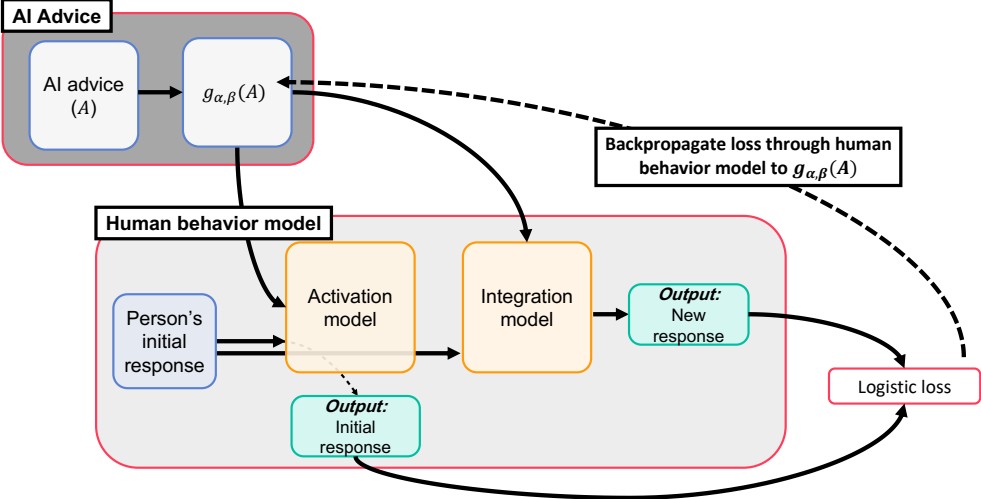

Figure 4: Overview of optimization procedure for $g_{\alpha,\beta}$. AI advice is input through $g_{\alpha,\beta}$. The transformed advice is then input into the human behavior model. Finally, loss is computed and the gradient with respect to the loss is backpropagated to update $\alpha$ and $\beta$. Note here that the human behavior model was trained previously and is now fixed.

Once we have optimized for $\alpha^*, \beta^*$, we can apply it to new data without needing the user demographics $\mathbf{u}$ (i.e., we do not personalize the confidence transformation to individual users). Note that the choice of loss function is arbitrary here (we used logistic loss as our task is binary classification). In particular, for high risk settings where a mistake is very costly, the $\alpha^*, \beta^*$ could be selected accordingly using an appropriate loss function.

Table 2: Simulated performance difference of modified advice ("Advice given" curve in Figure 3a) to unmodified advice. Positive difference indicates higher value for modified advice. Results shown across three average advice accuracy levels. The bolded row is the advice used in our human experiments.

| Advice Accuracy | Final Accuracy | Correct Confidence | Activation Rate |
|---|---|---|---|
| 75% | +0.7% | +0.097 | +0.083 |
| **79%** | **+1.9%** | **+0.114** | **+0.091** |
| 85% | +3.1% | +0.131 | +0.111 |

## 4.2 When does under/overconfident advice help?

Here we describe simple theoretical situations where uncalibrated advice improves performance. We consider a binary classification task and assume we want to minimize the logistic loss of the human-AI classifier. We also assume (1) a constrained version of the activation-integration model perfectly models human behavior; in particular, humans either use their initial response or follow the given advice exactly. And (2) we have an oracle for human behavior and know exactly the calibration of the human and AI advice. We will now construct a function $f(a, r_1)$ that outputs $a^*$, the optimal advice to minimize logistic loss of the human-AI system.

To select $a^*$, we need to consider two quantities:

$$L_{r_1} = -\left[p_{r_1} \log(r_1) + (1 - p_{r_1}) \log(1 - r_1)\right] \tag{3}$$
$$L_{a^*} = -\left[p_a \log(a^*) + (1 - p_a) \log(1 - a^*)\right] \tag{4}$$

Here, $p_a$ and $p_{r_1}$ are the probabilities the advice and initial response are correct respectively. Then $L_{r_1}$ relates the log-loss for non-activated humans and $L_{a^*}$ relates the log-loss of following the modified advice, $a^*$.

Now we simply select $a^*$ to minimize the expected loss incurred by the human-AI system

$$a^* = \arg\min_{a'} \ (1 - p_{\text{activation}})L_{r_1} + p_{\text{activation}}L_{a'} \tag{5}$$

for $p_{\text{activation}} = f_{\text{activation}}(a^*, r_1, \mathbf{u})$ (which we assumed we know exactly). For example, consider a setting where

$$p_{\text{activation}} = \mathbb{1}[\bar{r_1} + \epsilon < \bar{a}]$$
$$p_{r_1} = f(r_1)$$

where $\bar{x} = \max\{x, 1 - x\}$, $\epsilon \in [-0.5, 0.5]$, and for a generic function $f : [0, 1] \to [0, 1]$. Here a human under/overvalues their response relative to the advice (depending on $\epsilon$), and the human is not calibrated (depending on $f$). In this setting, it is indeed optimal for the advice, $a^*$, to be under/overconfident to correct for human misutilization of advice (depending on the advice and initial response). An example where $\epsilon = 0.1$ and $f(r_1) = r_1^2$ is shown in Figure 3b. Further discussion is included in Appendix C.

## 5 Results

### 5.1 Simulation results

Optimizing Equation 2 results in an optimal $\alpha^*, \beta^*$. Using $f_{HB}$, we can then analyze the expected benefit of $g_{\alpha^*, \beta^*}$ over the baseline, $g_{1,0}$. In Figure 3a, we show $g_{\alpha^*, \beta^*}$. We plot three separate curves. The curve labeled "Advice given" is optimized using the previously described dataset, and this is the function we use in our empirical results. We also synthetically modify the dataset by biasing the advice towards the correct label and adding noise to the advice to respectively increase and decrease the average advice accuracy in the dataset to 75% and 85%. We then fit a new $(\alpha, \beta)$ to the modified dataset and plot these curves. Note that $f_{HB}$ is fit to the unmodified dataset and is fixed for optimizing all three of these curves.

Table 3: Empirical performance on UK-based participants. Showing the difference in metric between modified advice and baseline (value is $> 0$ when modified advice resulted in a larger value). We recruited 50 participants for each task.

| Task | Final Accuracy | Correct Confidence | Activation Rate |
|---|---|---|---|
| Art (90%) | +8.5% | +0.150 | +0.101 |
| Cities (87%) | +1.7% | +0.140 | +0.169 |
| **All** (89%) | **+5.2%** | **+0.149** | **+0.143** |

For all three curves, we notice there is a large discontinuity at $A = 0.5$. This is a result of a large $\beta^*$, and suggests that advice is not useful to humans unless it is at a certain level of confidence (e.g., to make the human activated).

That all three curves follow relatively similar paths suggests the performance of $g_{\alpha^*, \beta^*}$ may be robust to small changes in advice accuracy. This is confirmed (in simulation) in Table 2, where we show the simulated performance difference between modified and unmodified advice. We see here that the final accuracy, confidence in correct result, and activation rate all increase with the modified advice.

## 5.2 Human experiments

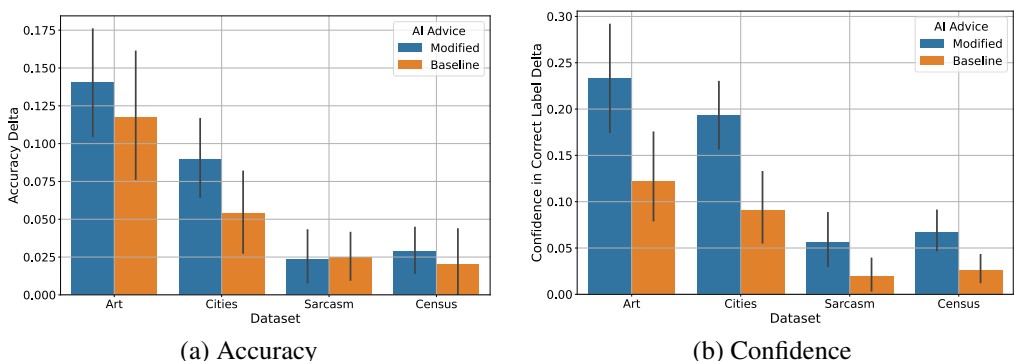

(a) Accuracy          (b) Confidence

Figure 5: Our empirical results: we plot the average change between the first (before advice) and second (after advice) responses in (a) accuracy and (b) confidence of the correct label, averaged by participant. These results are consistent with our simulation findings of Table 2. The error bar represents $\pm 1$ standard deviation.

We verify our simulation results across the four previously described tasks (with the exception that we do not add noise to the advice, so all participants see the exact same advice now). Adult, US-based participants were recruited through the Prolific crowdworker platform [23]. We recruited 50 participants for each task, and randomly assigned either the unmodified (baseline) or modified advice to each person. In Figure 5, we show the average change in (a) accuracy and (b) confidence of the correct label (the average confidence, with correct and incorrect labels receiving positive and negative weight respectively), partitioning by the advice received by participants. Accuracy and confidence increase across all tasks when using advice ($p = 3.84 \times 10^{-6}$). Moreover, we can largely confirm our simulation results. Accuracy and confidence increased due to the modified advice across nearly all tasks. Additionally, activation rate (not plotted) increased with the modified advice, with an average increase of 3.3% across tasks.

Another interesting finding is that tasks with higher advice accuracy (Art and Cities) exhibit larger increases in accuracy and confidence. The AI advice is more accurate in Art and Cities (accuracy: 90% and 87%) compared to Sarcasm and Census (accuracy: 78% for both). This intuitively makes sense – we are making the advice more confident in its predictions, and so while this benefits all tasks, it disproportionately benefits tasks where the advice is more accurate. This effect was predicted by our simulation results.

Our primary human experiments involved US-based participants. To show our findings generalize, we took the advice modification learned on the US data and applied it to new participants based in the UK for the Art and Cities tasks. The findings are highly consistent: modified advice increases activation and improves final human accuracy and confidence (Table 3). Combining US and UK data, performance improvement is significant for both accuracy ($p = 8.86 \times 10^{-3}$) and confidence ($p = 4.43 \times 10^{-11}$).

## 6  Discussion, Limitations, and Future Work

In this paper, we proposed optimizing a simple function to better calibrate AI advice for human use in a collaborative human-AI system. To optimize the reported confidence, we trained a human behavior model from experimental data and used it simulate the full human-AI system. The modified AI advice are explicitly uncalibrated, but in both simulated and empirical results, it resulted in substantially higher final human performance. We further validated our model using new participants from a different country. These results highlight the importance of recognizing that AI, when deployed in collaborative systems, should not be optimized in isolation. Our results also suggest that we can view modifying AI advice as a *nudge* for the human user. If the AI says its confidence is 0.7, the user may not fully absorb the import of 0.7 since humans are not great at intuitive probability. Modifying the reported confidence could be a useful nudge to help users to better absorb the information. This view is supported by prior work suggesting humans prefer clean decision boundaries [2].

Our work has several limitations that would be interesting to address in future studies. In particular, it is unclear how reliable such a method would be in practice. We argue that our method for creating *human-calibrated AI* can be deployed in practice, but requires an accurate model for human behavior. In particular, if we have access to an oracle for an individual's utilization of AI advice, we can exactly optimize the output of a calibrated AI model to correct for the individual's biases. Note that this may lead to under- or overcalibrated advice depending on the user and the advice confidence. However, as we cannot perfectly model any given individual, the question to ask is whether we can create a robust model for human behavior (e.g., with confidence intervals etc.) and then optimize with respect to that model while accounting for error. This extension was out-of-scope for this paper, and is an important direction of future research. Other limitations include: we considered tasks where advice accuracy generally exceeded individual human performance and did not evaluate expert tasks (e.g., involving clinicians making clinical decisions). We also did not consider how human interaction with an AI evolves over time – that is, we assumed each individual to be static over the course of the study, while in general, humans may naturally adapt to the AI and modifying the AI may affect this process. These are also important directions for future research. Finally, we recognize that the human behavior model we used in this work does not capture all real-world settings. It may be important to consider different models for different applications, and our results will need to be validated in these new settings.

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
