# OpenReview forum: "Uncalibrated Models Can Improve Human-AI Collaboration"
_NeurIPS.cc/2022/Conference — NeurIPS 2022 Accept_

### Official Review · Reviewer_k53k · 2022-07-06

**Rating:** 5
**Confidence:** 4
**Soundness:** 2 fair
**Presentation:** 3 good
**Contribution:** 3 good

**Summary:**

In the context of AI-assisted decision-making, the authors propose a method to adjust the confidence of the AI advice, that a human user observes on a given predictive task, with the goal to improve the user's accuracy on that task. For this purpose, they assume a two-stage cognitive mechanism for the user's processing of the reported AI confidence.  In the first stage, the users decide whether to use the advice or not, and if they do so, in the second stage, they decide to what extent they will use it. The authors fit a function for each stage using empirical data that they gathered on four binary classification settings. Then, they leverage these fitted functions to find which AI-confidence modification function--among a parametric family of functions-- minimizes the user's binary cross entropy loss on a specific training data set. Finally, the authors demonstrate  with simulations as well as experiments with real human users, that adjusting the AI confidence with such an optimized modification function, appears to improve not only the user's accuracy but also their confidence on their predictions.

**Questions:**

1. In the study the authors used prior aggregated human predictions as a proxy for the AI model, since the four datasets used in the experimental setting are too limited for training actual AI models. However, the sarcasm dataset as well as the census dataset are subsets of much larger datasets, namely the Reddit sarcasm dataset [1] and the Census dataset [2], that are suitable for actual model training.  Definitely, training (and calibrating)  models from scratch for the purposes of the study are beyond the scope of the paper, but given that the existing  state-of-the-art model [3] on the Reddit sarcasm dataset, reports F1 score > 70%, why such an off-the-shelf model, or similar ones were not used  as the AI model giving advice for the correspondent dataset?

2. In figure 2(b)  the initial response and advice are in $[-1,1]$ whereas in figures 3(b), and  9(b) they are in $[0,1]$. The authors do mention in line 196 that $r_1$ and $r_2$ are probabilities normalized in $[-1,1]$ and it would be helpful if the $r_1$ range was consistent across plots, or clearly specified otherwise.

3. The objective introduced in Eq. 2 is conditioned on the vector $\boldsymbol x$ of initial response $r_1$, advice A and user's features $u$. Also in Appendix C.4 it is mentioned that each advice-initial response pair was optimized separately. The objective between lines 249-250 refers to the expected value of the binary cross entropy loss on a given dataset, not conditioning on x or the advice-initial response pair. Is there an optimal $\alpha^*$, $\beta^*$ for each advice-initial response pair? Is there one for each $\boldsymbol x$ or there is one optimal $\alpha^*$, $\beta^*$ for a given dataset? From line 276, it seems that there is one $\alpha^*$, $\beta^*$ given a training dataset, though conditioning on x in Eq. (2) and stating in Appendix that optimization happens for each $(A,r_1)$ pair are rather confusing.

4. Minor typos:
       In line 533 "to modification" --> "no modification" ?
       In line 540 "for modify advice" --> "for modified advice"?



[1]  Mikhail Khodak, Nikunj Saunshi, and Kiran Vodrahalli. 2017. A large self-annotated corpus for sarcasm. arXiv preprint arXiv:1704.05579 (2017).

[2] Alex West and Anusha Praturu. 2019. Enhancing the Census Income Prediction Dataset. https://people.ischool.berkeley.edu/~alexwest/w210_census_income_html/.

[3] Taha Shangipour ataei, Soroush Javdan, and Behrouz Minaei-Bidgoli. 2020. Applying Transformers and Aspect-based Sentiment Analysis approaches on Sarcasm Detection. In Proceedings of the Second Workshop on Figurative Language Processing, pages 67–71, Online. Association for Computational Linguistics.

**Limitations:**

One key limitation to the proposed method is the need of accurately modeling human behavior, based on which the authors optimize the Ai's reported confidence with respect to the user's accuracy. The authors do acknowledge this limitation and clarify that computing such a human model  is not within the focus of their work. They also consider as future work a robust human model with which they could account for the model's error. Another limitation, that the authors consider as future work, is the fact that they used only an AI model with an accuracy higher than each user's accuracy, while not considering cases in which users' accuracy would exceed the model's accuracy. Furthermore, the proposed optimization is based on the choice of the training dataset D, which could be addressed as a limitation; it would be helpful if the authors could add some discussion or example of how a distribution shift on the test data could affect the user's performance or clarify if their method should only be applied in absence of such a shift.

**Strengths And Weaknesses:**

 The authors appear to be the first to make an attempt to manipulate the reported confidence of a decision-aid system, with the goal to maximize the user's accuracy when advised by such a system. Even though the idea that the reported system's confidence does affect the user's trust on the system's advice is not new and has been studied in prior works [1], [2], [3], the paper seems to be the first to formalize the problem of adjusting the reported confidence to increase the user's accuracy, opening an interesting new perspective for future works.

Overall the paper seems mostly well-structured and clearly written. There is a meticulous description on the experimental setting to gather user's data used to fit the activation and integration models as well as a detailed analysis of the models themselves. The formalization of the setting is also detailed, however the optimization objective and procedure could be described more clearly with more details, since for example conditioning on x on Eq.2, but not in equations between lines 249-250 confuses the reader. The authors provide theoretical examples, simulations and experimental results, with adequate analysis, based on which they evaluate their proposed method.

The idea of adjusting the reported confidence of an AI decision-aid system, with the goal to benefit the user's accuracy, as well as the experimental results do seem promising and opens different perspectives for future work. Overall, one might talk about an interesting and important first attempt to manipulate a system's reported confidence, to benefit the users.


[1] Sunita Sah, Don A Moore, and Robert J MacCoun. Cheap talk and credibility: The consequences of confidence and accuracy on advisor credibility and persuasiveness. Organizational Behavior and Human Decision Processes, 121(2):246–255, 2013.

[2] Zhang, Yunfeng, Q. Vera Liao, and Rachel KE Bellamy. "Effect of confidence and explanation on accuracy and trust calibration in AI-assisted decision making." In Proceedings of the 2020 Conference on Fairness, Accountability, and Transparency, pp. 295-305., 2020.

[3]Rechkemmer, Amy, and Ming Yin. "When Confidence Meets Accuracy: Exploring the Effects of Multiple Performance Indicators on Trust in Machine Learning Models." In CHI Conference on Human Factors in Computing Systems, pp. 1-14. 2022.

---

> ### Author Response · Authors · 2022-08-02
> **Thank you for your review!**
>
> Thank you for your thoughtful feedback! It’s very valuable in helping us to improve the paper.
>
> Questions —
>
> 1. We chose to use aggregated human predictions for all our tasks for uniformity in experiments. Additionally, for these experiments specifically, the source of the advice was not critical — we wanted calibrated advice that had a range of confidences across the questions asked. The aggregated advice met this criteria. Using an actual model, however, would be useful for future work (e.g., optimizing the model directly using our approach).
>
> 2. Thank you for your feedback here. We have adjusted the figures to ensure r1/r2 ranges are consistent throughout.
>
> 3. Thank you for pointing this out. This is a typo -- the conditional on x is correct. We have corrected this in the updated paper.
>
> 4. We have fixed the minor typos. Thank you for pointing these out.
>
>
> Question about optimal (alpha, beta) pair:
>
> In the way we formulated it, we chose to select a single (alpha, beta) pair optimized relative to the train dataset – that is, the (alpha, beta) that minimizes expected loss relative to the distribution of x observed in the training set. As you point out, we could have personalized the (alpha, beta) pair to individuals. This is a very interesting line of future work we are pursuing, but was out of scope for this paper.
>
>
> Question about distribution shift:
>
> This is a very interesting question. We believe our method (trained on a train dataset) would still apply under distribution shift given two assumptions:
>
> (1) the unmodified AI advice is still calibrated (or can be recalibrated) on the test distribution and
>
> (2) the human biases did not change on the test distribution (e.g., the human user can be more or less confident on the test data, but they should not change how they view the advice given to them on the test data).
>
> The intuition is that the goal of our method is to correct human biases related to the advice usage, and is agnostic to the actual confidence values / distribution of confidences on the test data.

---

> > ### Comment · Reviewer_k53k · 2022-08-09
> > **Thank you for your repsonse**
> >
> > Thank you for taking time to respond to all the questions and updating the paper.
> >
> > I still find a bit confusing conditioning on $\mathbf{x}$ and taking the sum over the $\mathbf{x}$ on a particular dataset between lines 246-247. If by conditioning on $\mathbf{x}$ you mean "minimize the expected loss relative to the distribution of x observed in the training set...", it might be useful to condition on the dataset instead or change the notation so that it is clear that the conditioning is not on a specific value of $\mathbf{x}$, otherwise from Eq. 2 one understands that you compute $\alpha^*,\beta^*$ for a particular value of $\mathbf{x}$.

---

### Official Review · Reviewer_LVB2 · 2022-07-10

**Rating:** 5
**Confidence:** 5
**Soundness:** 3 good
**Presentation:** 3 good
**Contribution:** 2 fair

**Summary:**

The paper proposes a method for adjusting model confidence such that the output model decision/recommendation it is best leveraged by a human. The method is based on the observation that humans may not be able to interpret confidence in the form of probabilistic estimate accurately, and may therefore prefer less calibrated notions of confidence. The authors show this via a set of user studies in 4 different domains. They use the data from these user studies to learn a model that re calibrates confidence depending human behavior, and they show that this benefits final team performance.

**Questions:**

My main questions are related to W1 (comparison to baselines) and W4 (what happens when the initial model is not calibrated).

Morevoer, there is an overall question on how these results would change when the model accuracy varies. One can assume that if the model has low accuracy, it will be less often useful to the user and gains in team performance will be lower. There could be a discussion around these in the limitations section.

**Limitations:**

The most important item to address here is the fact that depending on the cost\reward function of accurate and non-accurate decisions, uncalibration may lead to over reliance, which in the real world may have implications on safety.

**Strengths And Weaknesses:**

Strengths:

S1- The paper studies the problem of adjusting AI behavior specifically to improve human ai collaboration. This is a problem that deserves more attention in the community, especially for high-stake domains where the model is intended to augment a human and not operate autonomously.

S2- It verifies the observation on "uncalibration helping team performance" on several datasets and user studies.

S3- The method is practical to use and can even be used on an already trained model, assuming that that model is somewhat calibrated.

Weaknesses:

W1- The paper is missing comparisons with some important or obvious baselines. For example, comparing with a simple and pure step function (figure 3a) where the model only reveals its answer, this being an extreme case of uncalibration. Most importantly, there could have been a comparison with related work [1] mentioned in the literature. Technically, this is the analogous version of this paper which attempts to achieve similar results but during model optimization and not post hoc. That said, the relationship of the paper with [1] needs to be made more clear as the goals are really similar. The authors may also want to check the following paper [*] from the same authors, which provides evidence that humans prefer clean, non-stochastic error boundaries. Uncalibration as proposed by this work falls within the same framework, if the error boundaries are defined by using confidence.

[*] Beyond accuracy: The role of mental models in human-AI team performance

W2- Except [1], authors could also discuss post-hoc calibration in related work given that the proposed method is indeed a post-hoc calibration method for HAI. Note that the paper often claims for this to be an optimization approach but since it operates only on model confidences it is not such. This is not necessarily negative (in fact sometimes may be advantageous), but the discussion and framing of the approach needs to be made more precise.

W3- Depending on the utility function, uncalibration can be dangerous, if for example mistakes are associated with a much higher cost than the reward for correct predictions. This limitation needs to be discussed accordingly.

W4- The proposed approach may not work as well when the original confidence scores from the model are not calibrated. In this experiment they are reasonably calibrated since the model output is the majority vote of crowd answers. However, if this is not the case, the post-hoc calibration may not change much the situation or it may make it worse.

---

> ### Author Response · Authors · 2022-08-02
> **Thank you for your feedback!**
>
> Thank you for your thoughtful feedback, which has helped us to improve the paper! We greatly appreciate it.
>
> W1 —
> For the step function comparison: we actually did run experiments with this, and have added these results to the updated paper (Appendix G). Using a step function is worse than a continuous modification like the $g_{\alpha, \beta}$ we used in the main experiments. We hypothesize this is due to the step function removing any information about the quality of the advice (i.e., how confident the AI is from question to question), and so leads to less utilization of AI advice depending on the user’s perception.
>
> We added additional clarification on the relation of our work to [1] in lines 73-74,78-82. There are several recent works that have relations to ours, one of which is [1]. The key differences are summarized in lines 73-82:
>
> (1) We develop and use a model for human behavior from empirical data on human-AI interactions; in [1], they define a relatively simple model that is more convenient for analysis.
>
> (2) We demonstrate with empirical data (real human-AI interactions, not just in simulation) that human-AI performance improves with our method.
>
> (3) The setting is different. In our setting, everyone submits a response, views the advice, and resubmits a response. In [1], the advice is shown first, and the user has an (implicit) cost on whether to accept the advice or reject the advice and do the task themselves. These are both interesting settings, but they do have differences (e.g., people will be biased if they see the advice before attempting the task themselves).
>
> (4) Our perspective / goal is a bit different (it is related). In particular, we attempt to correct for biases in human usage of the AI (e.g., if a human underutilizes the advice purely because they devalue the AI, we attempt to correct for that). In [1], the authors are concerned with when people are willing to use advice (e.g., when it is highly confident), and seek to train the AI with this in mind (e.g., it is better to have a model with lower accuracy that is highly confident on a subset of the data than it is to have a higher accuracy but lower confidence on all the data).
>
> (5) We only require black-box access to the model, though our approach is generalizable for end-to-end learning. (Exploration of the end-to-end learning part was out of scope for this paper, though it is a very interesting follow-up.)
>
> Thank you for the additional reference [2]. We have added this reference in our discussion. We agree – the uncalibration does have a relation to the error boundaries described in [2], as our analysis suggests that making the advice uncalibrated makes it more clear when to use the advice / when the advice is better than the human user.
>
> [1] Gagan Bansal, Besmira Nushi, Ece Kamar, Eric Horvitz, and Daniel S Weld. Is the Most Accurate AI the Best Teammate? Optimizing AI for Teamwork.
> [2] Gagan Bansal, Besmira Nushi, Ece Kamar, Walter S. Lasecki, Daniel S. Weld, and Eric Horvitz. Beyond Accuracy: The Role of Mental Models in Human-AI Team Performance.
>
> W2 —
> We describe our method as an optimization approach because the framework we propose enables end-to-end optimization of the AI. In our experiments, we are appending a layer — the $g_{\alpha, \beta}$ function in our paper — to the model and optimizing its parameters. We could have optimized the last layer of a model as well, or indeed an entire model using our same approach (e.g., as a fine-tuning step). We chose to only optimize this $g_{\alpha,\beta}$ function in our experiments for simplicity and to keep the model as a black-box (and so can be thought of as a calibration method as you pointed out).
>
> W3 —
> Thank you for this suggestion. We discuss this in Section 4.2 and Appendix C when we give intuition for when uncalibrated advice can be beneficial to the user. In particular, when the uncalibrated advice is beneficial depends on the loss function and how it penalizes errors (and specifically, confident errors). We implicitly account for this during the selection procedure for alpha, beta. We added some more discussion in lines 251-252.
>
>
> W4 —
> We assumed the AI is already calibrated for simplicity. In section 4.2 and Appendix C, we give some intuition for why uncalibrated advice may be helpful. This analysis can be extended to include uncalibrated AI — we added some discussion in Appendix C.4. The key idea is that the optimal advice can be represented as a composition of (1) a function that calibrates the advice and (2) the modification function that uncalibrates the advice.
>
> If it is not possible to calibrate the advice, then we agree, modifying the advice may not make sense. In this case, it is unclear what the AI confidence is in the first place.

---

### Official Review · Reviewer_jiKw · 2022-07-12

**Rating:** 7
**Confidence:** 4
**Ethics Flag:** Yes
**Soundness:** 3 good
**Presentation:** 3 good
**Contribution:** 4 excellent

**Summary:**

This paper focuses on optimizing AI advice when AI is used as second opinion for a human decision maker. They propose a model for human interaction with AI that can be learned from data into two parts: an activation model and an integration model. They then provide a strategy to modify AI confidence given this human model to maximize task performance. They validate their results on four experimental tasks where they show that their approach of optimizing AI confidence increases performance compared to showing humans calibrated probabilities.

**Questions:**

“One potential question about our approach is: does modifying the AI’s confidence constitute misleading the user? We do not view this as misleading since a human’s assessment of confidence may differ from the statistical notion of calibrated probability.” – This is the most major point about this work. Even though I am positive about this paper, it is absolutely misleading the human and such approaches cannot be ethically implemented in practice as they are intentionally misleading. I believe even lay-people understand the notion

Why the two-stage interaction process instead of providing the AI advice from the beginning to mimic more real-world deployments?

What is baseline human accuracy and AI accuracy on all four tasks? Is there a baseline human-AI team, for example a learning to defer baseline, and results for that? It would be quite significant to show that humans deciding when to use AI advice is better than AI deciding when to use human advice.

Why weren’t alpha and beta optimized given the user demographics? Was there insufficient data? Furthermore, since alpha and beta weren’t personalized, then why does the activate/integrate functions need to be personalized?

The fact that “asks with higher advice accuracy (Art and Cities) exhibit larger increases in accuracy and confidence” is well documented in the literature since absolute AI advice accuracy is higher. Is there evidence of a relative increase in human accuracy (rather than absolute)?

Rather than misleading the users, what would you expect the effect of an intervention where users are taught to adapt their activation/integration of AI to revise their own internal model, could such an intervention mitigate the need for AI optimization and uncalibrated model?

What would you think the effect of telling users that one model is calibrated, and one is uncalibrated? For example, if you randomly split the user base that received the optimized advice, and told one part that the advice is calibrated (lie) and the other that it is uncalibrated (truth), what differences would you see?


**Ethics Review Area:**

["I don’t know"]

**Limitations:**

I think the authors should not encourage practice to adopt uncalibrated models, and should highlight that a limitation of their work, but still highlight that their work is an interesting exploration.

**Strengths And Weaknesses:**

Originality: The learning of the activation and integration models from human data are novel, even if the formulation is not as novel. However, the optimization procedure for AI advice is novel. Moreover, this is one of the first papers to do human-AI optimization directly.

Quality: The AI advice optimization procedure is technically sound. The experimental tasks are not very ecologically valid as they are not representative of tasks people will use AI for.

Clarity: The paper is well written. I think the human-AI optimization procedure can be better illustrated with a figure to show how $\alpha,\beta$ feed into both the activation and integration models. I would also include Figure 6 from the appendix in the main body.

Significance: To the best of my knowledge, this paper is one of the first to do AI optimization for AI assisted decision making and produce a positive result. I think this will inspire future work in this area, as I can see many extensions to this work. However, for practical significance, the authors need to strongly argue for why misleading users is okay, I strongly encourage them to reward the language of the intro and abstract as to have this paper be an academic exploration rather than a method AI designers should develop to improve human-AI teams in practice

---

> ### Author Response · Authors · 2022-08-02
> **Thank you for your helpful feedback!**
>
> Thank you very much for your helpful feedback and suggestions! We really appreciate it.
>
> We agree that the goal of this research is exploration of this interesting topic and that our findings should not be implemented as-is in practice. The main goal of the experiments was to highlight the importance of modeling the human decision making process in designing AI for human-AI collaboration. We have clarified this in the paper (lines 5-6, 15-16, 45-46).
>
> We also edited the language discussing how our modified AI can be misleading to human users (lines 50-54). We agree with you overall. It is an interesting question how calibrating AI’s answer as a nudge to the human user can lead the user to benefit more from the AI advise. We view this as asking “how should we present the AI output to a user to convey its meaning most usefully to that individual”. A full analysis would require a personalized modification function as well as knowledge about the human behavior model for the specific user (i.e., we need to know how an individual actually interprets the AI output to do this properly). This was out of scope for our paper, but will make for interesting future studies.
>
>
> Other questions / suggestions:
>
> - We added Figure 10 in the Appendix to better show the optimization process for alpha and beta.
>
> - The two-stage interaction process was chosen as it makes it easy to directly assess the effect of the advice. This two-stage process is common in psychology literature [1]. The two-stage process also mirrors how advice is commonly used in practical applications like medicine. Here, clinicians  generally seek advice (e.g., consulting a colleague,ordering a lab test to get more information, or consulting an AI advice giver) after their initial assessment [2].
>
> [1] Lyn M Van Swol, Jihyun Esther Paik, and Andrew Prahl. Advice recipients: The psychology of advice utilization.
>
> [2] Bunger, Alicia C., Nathan Doogan, Rochelle F. Hanson, and Sarah A. Birken. Advice-seeking during implementation: a network study of clinicians participating in a learning collaborative.
>
>
> - The baseline human accuracy is shown in Table 1 (in the column titled “Accuracy (before advice)”) and the AI advice accuracy is given in lines 303-304 (in the updated paper). We added a table in the Appendix where we repeat these values in one place (Appendix Section B.4, Table 4). We did not compare against other models for human-AI interaction, though we agree that would be very interesting to do. The purpose of this paper is to highlight that standard, calibrated AI are not optimal for human interaction.
>
> - We optimized a single set of alpha and beta parameters that we used for everyone. We opted to do this because it is the simplest and most direct approach. And as shown by our results, it was sufficient to demonstrate a performance gap between the modified and unmodified advice.
>
> The limitation is that it does not let us personalize the modification function to an individual (e.g., by optimizing the alpha, beta parameters using the person’s demographic information). While personalized modification functions is a very interesting idea, it was out of scope for this paper.
>
> - We used the participants’ provided information when training the activation and integration models as the goal was to create an accurate model for human behavior . To achieve this goal, we used all the information we could (including demographic information) to help the human behavior model better reflect the true behavior of the users in our train set.
>
> - Sorry, we’re not sure what you meant when you’re asking about a relative increase in human accuracy. Could you please clarify?
>
> - Your question about an intervention is a good idea. An intervention may indeed be sufficient as the purpose of the uncalibrated model is to correct for the internal biases a human has (which an intervention may also be able to accomplish). It is definitely worth exploring in future work.
>
>
> - For your question about telling users one model is calibrated and one model is uncalibrated — we might see one of two effects: (1) activation rate for the uncalibrated group to drop off significantly; telling people a negative about the AI (that it is uncalibrated) would affect usage of the advice or (2) little variation between the groups in the case that the distinction (calibrated vs. uncalibrated) is not properly understood by the user.

---

> > ### Comment · Reviewer_jiKw · 2022-08-06
> > **Thank you for the response**
> >
> > Thank you for clarifying my concerns.
> >
> > I would advise on placing Figure 10 in the main body of the paper.
> >
> > One question I have is why do you think the human+AI team did not outperform the AI accuracy on any of the tasks (please correct me if Im incorrect), and what should be done to overcome this in future work?
> >
> > The paper in it's current form I think is adequate for publication at NeurIPS.

---

### Official Review · Reviewer_4Md1 · 2022-07-13

**Rating:** 7
**Confidence:** 5
**Soundness:** 3 good
**Presentation:** 4 excellent
**Contribution:** 3 good

**Summary:**

The paper shows that calibrated models are not always the best for human-AI systems. Optimizing the ML model alone with human *uncalibrate* the model, but resulting in better performance. The paper provides rigorous experiments and related details.


**Questions:**

1. The paper shows that it's important to optimize the ML model with humans or potentially a model for human behaviors. It reminds me of the model-based RL problem. How does the reliability of the model for human behaviors affect the final human-AI performance?


**Suggestions**
1. Since the paper is related to human perceptions of the model's confidence, it will be great to have some human studies or interviews

**Limitations:**

The paper is easy to follow and the idea is well justified. As described in the last section, the experiment settings are limited and do not capture many real-world cases.

**Strengths And Weaknesses:**

**Strength**
1. The paper provides extensive and detailed experiment results that challenge the common belief of "better-calibrated models lead to better human-AI systems."


**Weakness**
1. The experiments are a little bit toy, with only two actions.

---

> ### Author Response · Authors · 2022-08-02
> **Thank you for your review!**
>
> Thank you for your helpful feedback! We are excited that you liked the paper.
>
> Regarding the choice of binary classification task. We chose the tasks we studied to be binary classification tasks as it makes studying the effect of varying advice sources much more tractable. Moreover, many settings where AI is applied involve binary decisions (e.g. anomaly detection, or some medical triage tasks). Additionally, we chose the tasks we used to ensure they would be doable by a layperson audience (i.e., people with no specific expertise would have moderate performance on all the tasks). While expert tasks are also interesting for practical use, it is much more difficult to recruit a large population of experts (e.g., it is significantly easier to recruit a large population of crowd-workers than it is to recruit a large population of doctors to perform a clinical decision task).
>
> Regarding your question on the reliability of the model. This is a good question — 2 observations we have:
>
> (1) The model we used is by no means a perfect model for human behavior, though it does have reasonable performance (Sections 3.1 and 3.2). However, even with this imperfect model, we see a significant increase in human-AI performance.
>
> (2) We have additional results that we have added to the supplement (Appendix G)  where we compare 3 versions of modified AI advice: 2 that are step functions and 1 that is similar to the modification function used in the main experiments. The main takeaway is that the step function seems to underperform relative to the $g_{\alpha, \beta}$ used in the main experiments. So, a poor-quality human behavior model (i.e., specifically, one that results in the step function modification being optimal), leads to worse performance.
>
>
> Taken together, these observations do suggest that human behavior model quality is important for our method. However, it is unclear whether there is a difference between a human behavior model of moderate quality vs. one of very high quality as we do not have access to such a model to test. This is a good direction for future studies.

---

### Meta-Review · Area_Chair_jq6o · 2022-08-24

**Recommendation:** Accept
**Confidence:** Certain

**Metareview:**

The paper studies the problem of optimizing an AI agent's reported confidence to improve the overall performance of a human user in a collaborative human-AI decision system. The main idea is to train a human behavior model based on experimental data that can be used to simulate the joint human-AI system and then optimize the AI agent's reported confidence based on this simulated system. The paper provides rigorous experimental evaluation involving human participants, and the results demonstrate the importance of jointly optimizing the human-AI system. The reviewers acknowledged that the paper considers an important problem and provides new insights on how to calibrate AI advice for human use. However, the reviewers also raised several concerns and questions in their initial reviews. We want to thank the authors for their detailed responses and for actively engaging with the reviewers during the discussion phase. The reviewers appreciated the responses, which helped in answering their key questions. The reviewers have an overall positive assessment of the paper, and there is a consensus for acceptance. The reviewers have provided detailed feedback in their reviews, and we strongly encourage the authors to incorporate this feedback when preparing the final version of the paper.

**Award:**

No

---

### Decision · Program_Chairs · 2022-09-14

Accept